# Sobolev Training for Neural Networks

**Wojciech Marian Czarnecki, Simon Osindero, Max Jaderberg**
**Grzegorz Swirszcz, and Razvan Pascanu**
DeepMind, London, UK
`{lejlot,osindero,jaderberg,swirszcz,razp}@google.com`

## Abstract

At the heart of deep learning we aim to use neural networks as function approximators – training them to produce outputs from inputs in emulation of a ground truth function or data creation process. In many cases we only have access to input-output pairs from the ground truth, however it is becoming more common to have access to derivatives of the target output with respect to the input – for example when the ground truth function is itself a neural network such as in network compression or distillation. Generally these target derivatives are not computed, or are ignored. This paper introduces Sobolev Training for neural networks, which is a method for incorporating these target derivatives in addition the to target values while training. By optimising neural networks to not only approximate the function's outputs but also the function's derivatives we encode additional information about the target function within the parameters of the neural network. Thereby we can improve the quality of our predictors, as well as the data-efficiency and generalization capabilities of our learned function approximation. We provide theoretical justifications for such an approach as well as examples of empirical evidence on three distinct domains: regression on classical optimisation datasets, distilling policies of an agent playing Atari, and on large-scale applications of synthetic gradients. In all three domains the use of Sobolev Training, employing target derivatives in addition to target values, results in models with higher accuracy and stronger generalisation.

## 1 Introduction

Deep Neural Networks (DNNs) are one of the main tools of modern machine learning. They are consistently proven to be powerful function approximators, able to model a wide variety of functional forms – from image recognition [8, 24], through audio synthesis [27], to human-beating policies in the ancient game of GO [22]. In many applications the process of training a neural network consists of receiving a dataset of input-output pairs from a ground truth function, and minimising some loss with respect to the network's parameters. This loss is usually designed to encourage the network to produce the same output, for a given input, as that from the target ground truth function. Many of the ground truth functions we care about in practice have an unknown analytic form, *e.g.* because they are the result of a natural physical process, and therefore we only have the observed input-output pairs for supervision. However, there are scenarios where we do know the analytic form and so are able to compute the ground truth gradients (or higher order derivatives), alternatively sometimes these quantities may be simply observable. A common example is when the ground truth function is itself a neural network; for instance this is the case for distillation [9, 20], compressing neural networks [7], and the prediction of synthetic gradients [12]. Additionally, if we are dealing with an environment/data-generation process (vs. a pre-determined set of data points), then even though we may be dealing with a black box we can still approximate derivatives using finite differences. In this work, we consider how this additional information can be incorporated in the learning process, and what advantages it can provide in terms of data efficiency and performance. We

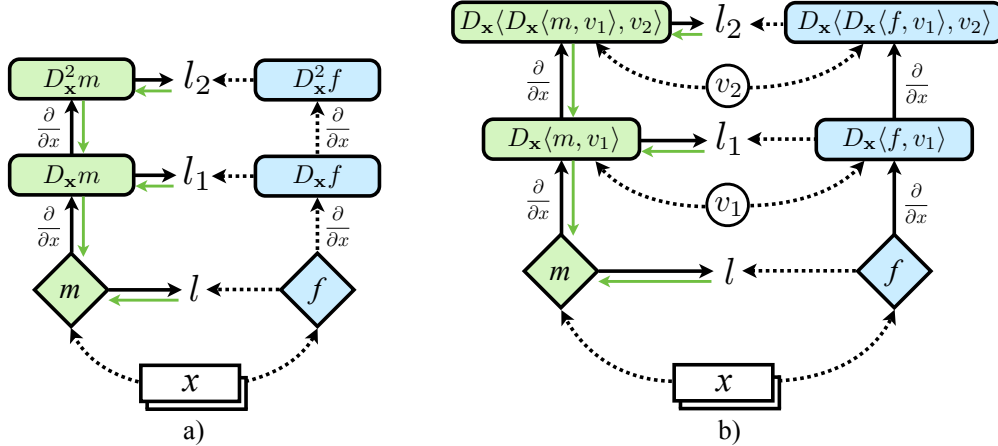

Figure 1: a) Sobolev Training of order 2. Diamond nodes $m$ and $f$ indicate parameterised functions, where $m$ is trained to approximate $f$. Green nodes receive supervision. Solid lines indicate connections through which error signal from loss $l$, $l_1$, and $l_2$ are backpropagated through to train $m$. b) Stochastic Sobolev Training of order 2. If $f$ and $m$ are multivariate functions, the gradients are Jacobian matrices. To avoid computing these high dimensional objects, we can efficiently compute and fit their projections on a random vector $v_j$ sampled from the unit sphere.

propose Sobolev Training (ST) for neural networks as a simple and efficient technique for leveraging derivative information about the desired function in a way that can easily be incorporated into any training pipeline using modern machine learning libraries.

The approach is inspired by the work of Hornik [10] which proved the universal approximation theorems for neural networks in Sobolev spaces – metric spaces where distances between functions are defined both in terms of their differences in values and differences in values of their derivatives.

In particular, it was shown that a sigmoid network can not only approximate a function's value arbitrarily well, but that the network's derivatives with respect to its inputs can approximate the corresponding derivatives of the ground truth function arbitrarily well too. Sobolev Training exploits this property, and tries to match not only the output of the function being trained but also its derivatives.

There are several related works which have also exploited derivative information for function approximation. For instance Wu et al. [30] and antecedents propose a technique for Bayesian optimisation with Gaussian Processess (GP), where it was demonstrated that the use of information about gradients and Hessians can improve the predictive power of GPs. In previous work on neural networks, derivatives of predictors have usually been used either to penalise model complexity (e.g. by pushing Jacobian norm to 0 [19]), or to encode additional, hand crafted invariances to some transformations (for instance, as in Tangentprop [23]), or estimated derivatives for dynamical systems [6] and very recently to provide additional learning signal during attention distillation [31][1]. Similar techniques have also been used in critic based Reinforcement Learning (RL), where a critic's derivatives are trained to match its target's derivatives [29, 15, 5, 4, 26] using small, sigmoid based models. Finally, Hyvärinen proposed Score Matching Networks [11], which are based on the somewhat surprising observation that one can model unknown derivatives of the function without actual access to its values – all that is needed is a sampling based strategy and specific penalty. However, such an estimator has a high variance [28], thus it is not really useful when true derivatives are given.

To the best of our knowledge and despite its simplicity, the proposal to directly match network derivatives to the true derivatives of the target function has been minimally explored for deep networks, especially modern ReLU based models. In our method, we show that by using the additional knowledge of derivatives with Sobolev Training we are able to train better models – models which achieve lower approximation errors and generalise to test data better – and reduce the sample complexity of learning. The contributions of our paper are therefore threefold: (**1**): We introduce

Sobolev Training – a new paradigm for training neural networks. (**2**): We look formally at the implications of matching derivatives, extending previous results of Hornik [10] and showing that modern architectures are well suited for such training regimes. (**3**): Empirical evidence demonstrating that Sobolev Training leads to improved performance and generalisation, particularly in low data regimes. Example domains are: regression on classical optimisation problems; policy distillation from RL agents trained on the Atari domain; and training deep, complex models using synthetic gradients – we report the first successful attempt to train a large-scale ImageNet model using synthetic gradients.

## 2 Sobolev Training

We begin by introducing the idea of training using Sobolev spaces. When learning a function $f$, we may have access to not only the output values $f(x_i)$ for training points $x_i$, but also the values of its $j$-th order derivatives with respect to the input, $D_{\mathbf{x}}^j f(x_i)$. In other words, instead of the typical training set consisting of pairs $\{(x_i, f(x_i))\}_{i=1}^N$ we have access to $(K+2)$-tuples $\{(x_i, f(x_i), D_{\mathbf{x}}^1 f(x_i), ..., D_{\mathbf{x}}^K f(x_i))\}_{i=1}^N$. In this situation, the derivative information can easily be incorporated into training a neural network model of $f$ by making derivatives of the neural network match the ones given by $f$.

Considering a neural network model $m$ parameterised with $\theta$, one typically seeks to minimise the empirical error in relation to $f$ according to some loss function $\ell$

$$\sum_{i=1}^N \ell(m(x_i|\theta), f(x_i)).$$

When learning in Sobolev spaces, this is replaced with:

$$\sum_{i=1}^N \left[ \ell(m(x_i|\theta), f(x_i)) + \sum_{j=1}^K \ell_j \left( D_{\mathbf{x}}^j m(x_i|\theta), D_{\mathbf{x}}^j f(x_i) \right) \right], \tag{1}$$

where $\ell_j$ are loss functions measuring error on $j$-th order derivatives. This causes the neural network to encode derivatives of the target function in its own derivatives. Such a model can still be trained using backpropagation and off-the-shelf optimisers.

A potential concern is that this optimisation might be expensive when either the output dimensionality of $f$ or the order $K$ are high, however one can reduce this cost through stochastic approximations. Specifically, if $f$ is a multivariate function, instead of a vector gradient, one ends up with a full Jacobian matrix which can be large. To avoid adding computational complexity to the training process, one can use an efficient, stochastic version of Sobolev Training: instead of computing a full Jacobian/Hessian, one just computes its projection onto a random vector (a direct application of a known estimation trick [19]). In practice, this means that during training we have a random variable $v$ sampled uniformly from the unit sphere, and we match these random projections instead:

$$\sum_{i=1}^N \left[ \ell(m(x_i|\theta), f(x_i)) + \sum_{j=1}^K \mathbb{E}_{v^j} \left[ \ell_j \left( \left\langle D_{\mathbf{x}}^j m(x_i|\theta), v^j \right\rangle, \left\langle D_{\mathbf{x}}^j f(x_i), v^j \right\rangle \right) \right] \right]. \tag{2}$$

Figure 1 illustrates compute graphs for non-stochastic and stochastic Sobolev Training of order 2.

## 3 Theory and motivation

While in the previous section we defined Sobolev Training, it is not obvious that modeling the derivatives of the target function $f$ is beneficial to function approximation, or that optimising such an objective is even feasible. In this section we motivate and explore these questions theoretically, showing that the Sobolev Training objective is a well posed one, and that incorporating derivative information has the potential to drastically reduce the sample complexity of learning.

Hornik showed [10] that neural networks with non-constant, bounded, continuous activation functions, with continuous derivatives up to order $K$ are universal approximators in the Sobolev spaces of order $K$, thus showing that sigmoid-networks are indeed capable of approximating elements of these

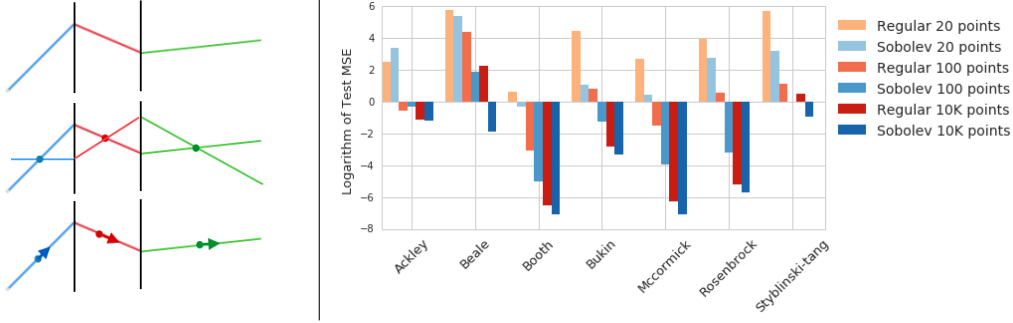

Figure 2: *Left:* From top: Example of the piece-wise linear function; Two (out of a continuum of) hypotheses consistent with 3 training points, showing that one needs two points to identify each linear segment; The only hypothesis consistent with 3 training points enriched with derivative information. *Right:* Logarithm of test error (MSE) for various optimisation benchmarks with varied training set size (20, 100 and 10000 points) sampled uniformly from the problem's domain.

spaces arbitrarily well. However, nowadays we often use activation functions such as ReLU which are neither bounded nor have continuous derivatives. The following theorem shows that for $K = 1$ we can use ReLU function (or a similar one, like leaky ReLU) to create neural networks that are universal approximators in Sobolev spaces. We will use a standard symbol $\mathcal{C}^1(S)$ (or simply $\mathcal{C}^1$) to denote a space of functions which are continuous, differentiable, and have a continuous derivative on a space $S$ [14]. All proofs are given in the Supplementary Materials (SM).

**Theorem 1.** *Let $f$ be a $\mathcal{C}^1$ function on a compact set. Then, for every positive $\varepsilon$ there exists a single hidden layer neural network with a ReLU (or a leaky ReLU) activation which approximates $f$ in Sobolev space $\mathcal{S}_1$ up to $\epsilon$ error.*

This suggests that the Sobolev Training objective is achievable, and that we can seek to encode the values and derivatives of the target function in the values and derivatives of a ReLU neural network model. Interestingly, we can show that if we seek to encode an arbitrary function in the derivatives of the model then this is impossible not only for neural networks but also for any arbitrary differentiable predictor on compact sets.

**Theorem 2.** *Let $f$ be a $\mathcal{C}^1$ function. Let $g$ be a continuous function satisfying $\|g - \frac{\partial f}{\partial x}\|_\infty > 0$. Then, there exists an $\eta > 0$ such that for any $\mathcal{C}^1$ function $h$ either $\|f - h\|_\infty \geq \eta$ or $\left\|g - \frac{\partial h}{\partial x}\right\|_\infty \geq \eta$.*

However, when we move to the regime of finite training data, we can encode any arbitrary function in the derivatives (as well as higher order signals if the resulting Sobolev spaces are not degenerate), as shown in the following Proposition.

**Proposition 1.** *Given any two functions $f : S \rightarrow \mathbb{R}$ and $g : S \rightarrow \mathbb{R}^d$ on $S \subseteq \mathbb{R}^d$ and a finite set $\Sigma \subset S$, there exists neural network $h$ with a ReLU (or a leaky ReLU) activation such that $\forall x \in \Sigma : f(x) = h(x)$ and $g(x) = \frac{\partial h}{\partial x}(x)$ (it has 0 training loss).*

Having shown that it is possible to train neural networks to encode both the values and derivatives of a target function, we now formalise one possible way of showing that Sobolev Training has lower sample complexity than regular training.

Let $\mathcal{F}$ denote the family of functions parametrised by $\omega$. We define $K_{reg} = K_{reg}(\mathcal{F})$ to be a measure of the amount of data needed to learn some target function $f$. That is $K_{reg}$ is the smallest number for which there holds: for every $f_\omega \in \mathcal{F}$ and every set of distinct $K_{reg}$ points $(x_1, ..., x_{K_{reg}})$ such that $\forall_{i=1,...,K_{reg}} f(x_i) = f_\omega(x_i) \Rightarrow f = f_\omega$. $K_{sob}$ is defined analogously, but the final implication is of form $f(x_i) = f_\omega(x_i) \wedge \frac{\partial f}{\partial x}(x_i) = \frac{\partial f_\omega}{\partial x}(x_i) \Rightarrow f = f_\omega$. Straight from the definition there follows:

**Proposition 2.** *For any $\mathcal{F}$, there holds $K_{sob}(\mathcal{F}) \leq K_{reg}(\mathcal{F})$.*

For many families, the above inequality becomes sharp. For example, to determine the coefficients of a polynomial of degree $n$ one needs to compute its values in at least $n + 1$ distinct points. If we know values and the derivatives at $k$ points, it is a well-known fact that only $\lceil \frac{n}{2} \rceil$ points suffice to determine all the coefficients. We present two more examples in a slightly more formal way. Let $\mathcal{F}_G$ denote a family of Gaussian PDF-s (parametrised by $\mu$, $\sigma$). Let $\mathbb{R}^d \supset D = D_1 \cup \ldots \cup D_n$ and let $\mathcal{F}_{PL}$ be a family of functions from $D_1 \times ... \times D_n$ (Cartesian product of sets $D_i$) to $\mathbb{R}^n$ of form $f(x) = [A_1 x_1 + b_1, \ldots, A_n x_n + b_n]$ (linear element-wise) (Figure 2 Left).

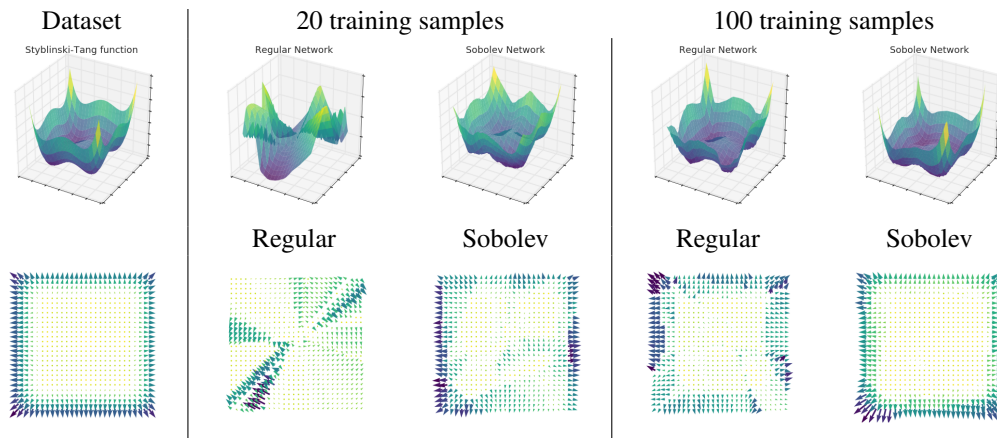

Figure 3: Styblinski-Tang function (on the left) and its models using regular neural network training (left part of each plot) and Sobolev Training (right part). We also plot the vector field of the gradients of each predictor underneath the function plot.

**Proposition 3.** *There holds $K_{sob}(\mathcal{F}_\mathrm{G}) < K_{reg}(\mathcal{F}_\mathrm{G})$ and $K_{sob}(\mathcal{F}_\mathrm{PL}) < K_{reg}(\mathcal{F}_\mathrm{PL})$.*

This result relates to Deep ReLU networks as they build a hyperplanes-based model of the target function. If those were parametrised independently one could expect a reduction of sample complexity by $d+1$ times, where $d$ is the dimension of the function domain. In practice parameters of hyperplanes in such networks are not independent, furthermore the hinges positions change so the Proposition cannot be directly applied, but it can be seen as an intuitive way to see why the sample complexity drops significantly for Deep ReLU networks too.

## 4 Experimental Results

We consider three domains where information about derivatives is available during training[2].

### 4.1 Artificial Data

First, we consider the task of regression on a set of well known low-dimensional functions used for benchmarking optimisation methods.

We train two hidden layer neural networks with 256 hidden units per layer with ReLU activations to regress towards function values, and verify generalisation capabilities by evaluating the mean squared error on a hold-out test set. Since the task is standard regression, we choose all the losses of Sobolev Training to be L2 errors, and use a first order Sobolev method (second order derivatives of ReLU networks with a linear output layer are constant, zero). The optimisation is therefore:

$$\min_\theta \frac{1}{N} \sum_{i=1}^{N} \|f(x_i) - m(x_i|\theta)\|_2^2 + \|\nabla_x f(x_i) - \nabla_x m(x_i|\theta)\|_2^2.$$

Figure 2 right shows the results for the optimisation benchmarks. As expected, Sobolev trained networks perform extremely well – for six out of seven benchmark problems they significantly reduce the testing error with the obtained errors orders of magnitude smaller than the corresponding errors of the regularly trained networks. The stark difference in approximation error is highlighted in Figure 3, where we show the Styblinski-Tang function and its approximations with both regular and Sobolev Training. It is clear that even in very low data regimes, the Sobolev trained networks can capture the functional shape.

Looking at the results, we make two important observations. First, the effect of Sobolev Training is stronger in low-data regimes, however it does not disappear even in the high data regime, when one has 10,000 training examples for training a two-dimensional function. Second, the only case where regular regression performed better is the regression towards Ackley's function. This particular

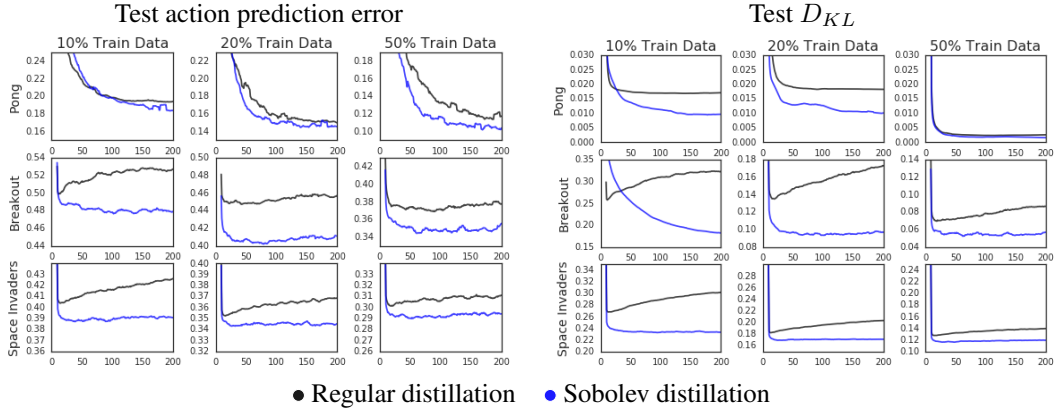

<div align="center">● Regular distillation　　● Sobolev distillation</div>

Figure 4: Test results of distillation of RL agents on three Atari games. Reported test action prediction error (left) is the error of the most probable action predicted between the distilled policy and target policy, and test $D_{KL}$ (right) is the Kulblack-Leibler divergence between policies. Numbers in the column title represents the percentage of the 100K recorded states used for training (the remaining are used for testing). In all scenarios the Sobolev distilled networks are significantly more similar to the target policy.

example was chosen to show that one possible weak point of our approach might be approximating functions with a very high frequency signal component in the relatively low data regime. Ackley's function is composed of exponents of high frequency cosine waves, thus creating an extremely bumpy surface, consequently a method that tries to match the derivatives can behave badly during testing if one does not have enough data to capture this complexity. However, once we have enough training data points, Sobolev trained networks are able to approximate this function better.

## 4.2   Distillation

Another possible application of Sobolev Training is to perform model distillation. This technique has many applications, such as network compression [21], ensemble merging [9], or more recently policy distillation in reinforcement learning [20].

We focus here on a task of distilling a policy. We aim to distill a target policy $\pi^*(s)$ – a trained neural network which outputs a probability distribution over actions – into a smaller neural network $\pi(s|\theta)$, such that the two policies $\pi^*$ and $\pi$ have the same behaviour. In practice this is often done by minimising an expected divergence measure between $\pi^*$ and $\pi$, for example, the Kullback–Leibler divergence $D_{KL}(\pi(s)\|\pi^*(s))$, over states gathered while following $\pi^*$. Since policies are multivariate functions, direct application of Sobolev Training would mean producing full Jacobian matrices with respect to the $s$, which for large actions spaces is computationally expensive. To avoid this issue we employ a stochastic approximation described in Section 2, thus resulting in the objective

$$\min_\theta D_{KL}(\pi(s|\theta)\|\pi^*(s)) + \alpha \mathbb{E}_v \left[ \| \nabla_s \langle \log \pi^*(s), v \rangle - \nabla_s \langle \log \pi(s|\theta), v \rangle \| \right],$$

where the expectation is taken with respect to $v$ coming from a uniform distribution over the unit sphere, and Monte Carlo sampling is used to approximate it.

As target policies $\pi^*$, we use agents playing Atari games [17] that have been trained with A3C [16] on three well known games: Pong, Breakout and Space Invaders. The agent's policy is a neural network consisting of 3 layers of convolutions followed by two fully-connected layers, which we distill to a smaller network with 2 convolutional layers and a single smaller fully-connected layer (see SM for details). Distillation is treated here as a purely supervised learning problem, as our aim is not to re-evaluate known distillation techniques, but rather to show that if the aim is to minimise a given divergence measure, we can improve distillation using Sobolev Training. Figure 4 shows test error during training with and without Sobolev Training[3]. The introduction of Sobolev Training leads to similar effects as in the previous section – the network generalises much more effectively, and this

Table 1: Various techniques for producing synthetic gradients. Green shaded nodes denote nodes that get supervision from the corresponding object from the main network (gradient or loss value). We report accuracy on the test set $\pm$ standard deviation. Backpropagation results are given in parenthesis.

| | Noprop | Direct SG [12] | VFBN [25] | Critic | Sobolev |
|---|---|---|---|---|---|
| **CIFAR-10 with 3 synthetic gradient modules** | | | | | |
| Top 1 (94.3%) | 54.5% $\pm_{1.15}$ | 79.2% $\pm_{0.01}$ | 88.5% $\pm_{2.70}$ | 93.2% $\pm_{0.02}$ | 93.5% $\pm_{0.01}$ |
| **ImageNet with 1 synthetic gradient module** | | | | | |
| Top 1 (75.0%) | 54.0% $\pm_{0.29}$ | - | 57.9% $\pm_{2.03}$ | 71.7% $\pm_{0.23}$ | 72.0% $\pm_{0.05}$ |
| Top 5 (92.3%) | 77.3% $\pm_{0.06}$ | - | 81.5% $\pm_{1.20}$ | 90.5% $\pm_{0.15}$ | 90.8% $\pm_{0.01}$ |
| **ImageNet with 3 synthetic gradient modules** | | | | | |
| Top 1 (75.0%) | 18.7% $\pm_{0.18}$ | - | 28.3% $\pm_{5.24}$ | 65.7% $\pm_{0.56}$ | 66.5% $\pm_{0.22}$ |
| Top 5 (92.3%) | 38.0% $\pm_{0.34}$ | - | 52.9% $\pm_{6.62}$ | 86.9% $\pm_{0.33}$ | 87.4% $\pm_{0.11}$ |

is especially true in low data regimes. Note the performance gap on Pong is small due to the fact that optimal policy is quite degenerate for this game[4]. In all remaining games one can see a significant performance increase from using our proposed method, and as well as minor to no overfitting.

Despite looking like a regularisation effect, we stress that Sobolev Training is not trying to find the simplest models for data or suppress the expressivity of the model. This training method aims at matching the original function's smoothness/complexity and so reduces overfitting by effectively extending the information content of the training set, rather than by imposing a data-independent prior as with regularisation.

### 4.3 Synthetic Gradients

The previous experiments have shown how information about the derivatives can boost approximating function values. However, the core idea of Sobolev Training is broader than that, and can be employed in both directions. Namely, if one ultimately cares about approximating derivatives, then additionally approximating values can help this process too. One recent technique, which requires a model of gradients is Synthetic Gradients (SG) [12] – a method for training complex neural networks in a decoupled, asynchronous fashion. In this section we show how we can use Sobolev Training for SG.

The principle behind SG is that instead of doing full backpropagation using the chain-rule, one splits a network into two (or more) parts, and approximates partial derivatives of the loss $L$ with respect to some hidden layer activations $h$ with a trainable function $SG(h, y|\theta)$. In other words, given that network parameters up to $h$ are denoted by $\Theta$

$$\frac{\partial L}{\partial \Theta} = \frac{\partial L}{\partial h}\frac{\partial h}{\partial \Theta} \approx SG(h, y|\theta)\frac{\partial h}{\partial \Theta}.$$

In the original SG paper, this module is trained to minimise $L_{SG}(\theta) = \left\| SG(h, y|\theta) - \frac{\partial L(p_h, y)}{\partial h} \right\|_2^2$, where $p_h$ is the final prediction of the main network for hidden activations $h$. For the case of learning a classifier, in order to apply Sobolev Training in this context we construct a loss predictor, composed

of a class predictor $p(\cdot|\theta)$ followed by the log loss, which gets supervision from the true loss, and the gradient of the prediction gets supervision from the true gradient:

$$m(h,y|\theta) := L(p(h|\theta),y), \quad SG(h,y|\theta) := \partial m(h,y|\theta)/\partial h,$$

$$L_{SG}^{sob}(\theta) = \ell(m(h,y|\theta), L(p_h,y))) + \ell_1\left(\frac{\partial m(h,y|\theta)}{\partial h}, \frac{\partial L(p_h,y)}{\partial h}\right).$$

In the Sobolev Training framework, the target function is the loss of the main network $L(p_h,y)$ for which we train a model $m(h,y|\theta)$ to approximate, and in addition ensure that the model's derivatives $\partial m(h,y|\theta)/\partial h$ are matched to the true derivatives $\partial L(p_h,y)/\partial h$. The model's derivatives $\partial m(h,y|\theta)/\partial h$ are used as the synthetic gradient to decouple the main network.

This setting closely resembles what is known in reinforcement learning as critic methods [13]. In particular, if we do not provide supervision on the gradient part, we end up with a loss critic. Similarly if we do not provide supervision at the loss level, but only on the gradient component, we end up in a method that resembles VFBN [25]. In light of these connections, our approach in this application setting can be seen as a generalisation and unification of several existing ones (see Table 1 for illustrations of these approaches).

One could ask why we need these additional constraints, and what is gained over using a neural network based approximator directly [12]. The answer lies in the fact that gradient vector fields are a tiny subset of all vector fields, and while each neural network produces a valid vector field, almost no (standard) neural network produces valid gradient vector fields. Using non-gradient vector fields as update directions for learning can have catastrophic consequences – learning divergence, oscillations, chaotic behaviour, etc. The following proposition makes this observation more formal:

**Proposition 4.** *If an approximator $SG(h,y|\theta)$ produces a valid gradient vector field of some scalar function L then the approximator's Jacobian matrix must be symmetric.*

It is worth noting that having a symmetric Jacobian is an extremely rare property for a neural network model. For example, a linear model has a symmetric Jacobian if and only if its weight matrix is symmetric. If we sample weights iid from typical distribution (like Gaussian or uniform on an interval), the probability of sampling such a matrix is 0, but it could be easy to learn with strong, symmetric-enforcing updates. On the other hand, for highly non-linear neural networks, it is not only improbable to randomly find such a model, but enforcing this constraint during learning becomes much harder too. This might be one of the reasons why linear SG modules work well in Jaderberg et al. [12], but non-linear convolutional SG struggled to achieve state-of-the-art performance.

When using Sobolev-like approach SG always produces a valid gradient vector field by construction, thus avoiding the problem described.

We perform experiments on decoupling deep convolutional neural network image classifiers using synthetic gradients produced by loss critics that are trained with Sobolev Training, and compare to regular loss critic training, and regular synthetic gradient training. We report results on CIFAR-10 for three network splits (and therefore three synthetic gradient modules) and on ImageNet with one and three network splits [5].

The results are shown in Table 1. With a naive SG model, we obtain 79.2% test accuracy on CIFAR-10. Using an SG architecture which resembles a small version of the rest of the model makes learning much easier and led to 88.5% accuracy, while Sobolev Training achieves 93.5% final performance. The regular critic also trains well, achieving 93.2%, as the critic forces the lower part of the network to provide a representation which it can use to reduce the classification (and not just prediction) error. Consequently it provides a learning signal which is well aligned with the main optimisation. However, this can lead to building representations which are suboptimal for the rest of the network. Adding additional gradient supervision by constructing our Sobolev SG module avoids this issue by making sure that synthetic gradients are truly aligned and gives an additional boost to the final accuracy.

For ImageNet [3] experiments based on ResNet50 [8], we obtain qualitatively similar results. Due to the complexity of the model and an almost 40% gap between no backpropagation and full backpropagation results, the difference between methods with vs without loss supervision grows significantly. This suggests that at least for ResNet-like architectures, loss supervision is a crucial

component of a SG module. After splitting ResNet50 into four parts the Sobolev SG achieves 87.4% top 5 accuracy, while the regular critic SG achieves 86.9%, confirming our claim about suboptimal representation being enforced by gradients from a regular critic. Sobolev Training results were also much more reliable in all experiments (significantly smaller standard deviation of the results).

## 5    Discussion and Conclusion

In this paper we have introduced Sobolev Training for neural networks – a simple and effective way of incorporating knowledge about derivatives of a target function into the training of a neural network function approximator. We provided theoretical justification that encoding both a target function's value as well as its derivatives within a ReLU neural network is possible, and that this results in more data efficient learning. Additionally, we show that our proposal can be efficiently trained using stochastic approximations if computationally expensive Jacobians or Hessians are encountered.

In addition to toy experiments which validate our theoretical claims, we performed experiments to highlight two very promising areas of applications for such models: one being distillation/compression of models; the other being the application to various meta-optimisation techniques that build models of other models dynamics (such as synthetic gradients, learning-to-learn, etc.). In both cases we obtain significant improvement over classical techniques, and we believe there are many other application domains in which our proposal should give a solid performance boost.

In this work we focused on encoding true derivatives in the corresponding ones of the neural network. Another possibility for future work is to encode information which one believes to be highly correlated with derivatives. For example curvature [18] is believed to be connected to uncertainty. Therefore, given a problem with known uncertainty at training points, one could use Sobolev Training to match the second order signal to the provided uncertainty signal. Finite differences can also be used to approximate gradients for black box target functions, which could help when, for example, learning a generative temporal model. Another unexplored path would be to apply Sobolev Training to *internal derivatives* rather than just derivatives with respect to the inputs.

## Footnotes

[1]Please relate to Supplementary Materials, section 5 for details

[2]All experiments were performed using TensorFlow [2] and the Sonnet neural network library [1].

[3]Testing is performed on a held out set of episodes, thus there are no temporal nor causal relations between training and testing

[4]For majority of the time the policy in Pong is uniform, since actions taken when the ball is far away from the player do not matter at all. Only in crucial situations it peaks so the ball hits the paddle.

[5]N.b. the experiments presented use learning rates, annealing schedule, etc. optimised to maximise the backpropagation baseline, rather than the synthetic gradient decoupled result (details in the SM).

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
