[Supplementary Material]

# Supplementary Materials for "Sobolev Training for Neural Networks"

## 1 Proofs

**Theorem 1.** *Let $f$ be a $\mathcal{C}^1$ function on a compact set. Then, for every positive $\varepsilon$ there exists a single hidden layer neural network with a ReLU (or a leaky ReLU) activation which approximates $f$ in Sobolev space $\mathcal{S}_1$ up to $\epsilon$ error.*

We start with a definition. We will say that a function $p$ on a set $D$ is *piecewise-linear*, if there exist $D_1, \ldots, D_n$ such that $D = D_1 \cup \ldots \cup D_n = D$ and $p|_{D_i}$ is linear for every $i = 1, \ldots, n$ (note, that we assume finiteness in the definition).

**Lemma 1.** *Let $D$ be a compact subset of $\mathbb{R}$ and let $\varphi \in \mathcal{C}^1(D)$. Then, for every $\varepsilon > 0$ there exists a piecewise-linear, continuous function $p : D \to \mathbb{R}$ such that $|\varphi(x) - p(x)| < \varepsilon$ for every $x \in D$ and $|\varphi'(x) - p'(x)| < \varepsilon$ for every $x \in D \setminus P$, where $P$ is the set of points of non-differentiability of $p$.*

*Proof.* By assumption, the function $\varphi'$ is continuous on $D$. Every continuous function on a compact set has to be uniformly continuous. Therefore, there exists $\delta_1$ such that for every $x_1$, $x_2$, with $|x_1 - x_2| < \delta_1$ there holds $|\varphi'(x_1) - \varphi'(x_2)| < \varepsilon$. Moreover, $\varphi'$ has to be bounded. Let $M$ denote $\sup_x |\varphi'(x)|$. By Mean Value Theorem, if $|x_1 - x_2| < \frac{\varepsilon}{2M}$ then $|\varphi(x_1) - \varphi(x_2)| < \frac{\varepsilon}{2}$. Let $\delta = \min \left\{ \delta_1, \frac{\varepsilon}{2M} \right\}$. Let $\xi_i$, $i = 0, \ldots, N$ be a sequence satisfying: $\xi_i < \xi_j$ for $i < j$, $|\xi_i - \xi_{i-1}| < \delta$ for $i = 1, \ldots, N$ and $\xi_0 < x < \xi_N$ for all $x \in D$. Such sequence obviously exists, because $D$ is a compact (and thus bounded) subset of $\mathbb{R}$. We define

$$p(x) = \varphi(\xi_{i-1}) + \frac{\varphi(\xi_i) - \varphi(\xi_{i-1})}{\xi_i - \xi_{i-1}}(x - \xi_{i-1}) \quad \text{for} \quad x \in [\xi_{i-1}, \xi_i] \cap D.$$

It can be easily verified, that it has all the desired properties. Indeed, let $x \in D$. Let $i$ be such that $\xi_{i-1} \leq x \leq \xi_i$. Then $|\varphi(x) - p(x)| = |\varphi(x) - \varphi(\xi_i) + p(\xi_i) - p(x)| \leq |\varphi(x) - \varphi(\xi_i)| + |p(\xi_i) - p(x)| \leq \varepsilon$, as $\varphi(\xi_i) = p(\xi_i)$ and $|\xi_i - x| \leq |\xi_i - \xi_{i-1}| < \delta$ by definitions. Moreover, applying Mean Value Theorem we get that there exists $\zeta \in [\xi_{i-1}, \xi_i]$ such that $\varphi'(\zeta) = \frac{\varphi(\xi_i) - \varphi(\xi_{i-1})}{\xi_i - \xi_{i-1}} = p'(\zeta)$. Thus, $|\varphi'(x) - p'(x)| = |\varphi'(x) - \varphi'(\zeta) + p'(\zeta) - p'(x)| \leq |\varphi'(x) - \varphi'(\zeta)| + |p'(\zeta) - p'(x)| \leq \varepsilon$ as $p'(\zeta) = p'(x)$ and $|\zeta - x| < \delta$. $\square$

**Lemma 2.** *Let $\varphi \in \mathcal{C}^1(\mathbb{R})$ have finite limits $\lim_{x \to -\infty} \varphi(x) = \varphi_-$ and $\lim_{x \to \infty} \varphi(x) = \varphi_+$, and let $\lim_{x \to -\infty} \varphi'(x) = \lim_{x \to \infty} \varphi'(x) = 0$. Then, for every $\varepsilon > 0$ there exists a piecewise-linear, continuous function $p : \mathbb{R} \to \mathbb{R}$ such that $|\varphi(x) - p(x)| < \varepsilon$ for every $x \in \mathbb{R}$ and $|\varphi'(x) - p'(x)| < \varepsilon$ for every $x \in \mathbb{R} \setminus P$, where $P$ is the set of points of non-differentiability of $p$.*

*Proof.* By definition of a limit there exist numbers $K_- < K_+$ such that $x < K_- \Rightarrow |\varphi(x) - \varphi_-| \leq \frac{\varepsilon}{2}$ and $x > K_+ \Rightarrow |\varphi(x) - \varphi_+| \leq \frac{\varepsilon}{2}$. We apply Lemma 1 to the function $\varphi$ and the set $D = [K, K_+]$. We define $\tilde{p}$ on $[K_-, K_+]$ according to Lemma 1. We define $p$ as

$$p(x) = \begin{cases} \varphi_- & \text{for} \quad x \in [-\infty, K_-] \\ \tilde{p}(x) & \text{for} \quad x \in [K_-, K_+] \\ \varphi_+ & \text{for} \quad x \in [K_+, \infty] \end{cases}.$$

It can be easily verified, that it has all the desired properties. $\square$

**Corollary 1.** *For every $\varepsilon > 0$ there exists a combination of ReLU functions which approximates a sigmoid function with accurracy $\varepsilon$ in the Sobolev space.*

*Proof.* It follows immediately from Lemma 2 and the fact, that any piecewise-continuous function on $\mathbb{R}$ can be expressed as a finite sum of ReLU activations. $\square$

**Remark 1.** *The authors decided, for the sake of clarity and better readability of the paper, to not treat the issue of non-differentiabilities of the piecewise-linear function at the junction points. It can be approached in various ways, either by noticing they form a finite, and thus a zero-Lebesgue measure set and invoking the formal definition f Sobolev spaces, or by extending the definition of a derivative, but it leads only to non-interesting technical complications.*

*Proof of Theorem 1.* By Hornik's result (Hornik [10]) there exists a combination of $N$ sigmoids approximating the function $f$ in the Sobolev space with $\frac{\varepsilon}{2}$ accuracy. Each of those sigmoids can, in turn, be approximated up to $\frac{\varepsilon}{2N}$ accuracy by a finite combination of ReLU (or leaky ReLU) functions (Corollary 1), and the theorem follows. $\square$

**Theorem 2.** *Let $f$ be a $\mathcal{C}^1(S)$. Let $g$ be a continuous function satisfying $\|g - \frac{\partial f}{\partial x}\| > 0$. Then, there exists an $\varepsilon = \varepsilon(f, g)$ such that for any $\mathcal{C}^1$ function $h$ there holds either $\|f - h\| \geq \varepsilon$ or $\|g - \frac{\partial h}{\partial x}\| \geq \varepsilon$.*

*Proof.* Assume that the converse holds. This would imply, that there exists a sequence of functions $h_n$ such that $\lim\limits_{n\to\infty} \frac{\partial h_n}{\partial x} = g$ and $\lim\limits_{n\to\infty} h_n = f$. A theorem about term-by-term differentiation implies then that the limit $\lim\limits_{n\to\infty} h_n$ is differentiable, and that the equality $\frac{\partial}{\partial x}\left( \lim\limits_{n\to\infty} h_n \right) = \frac{\partial f}{\partial x}$ holds. However, $\frac{\partial}{\partial x}\left( \lim\limits_{n\to\infty} h_n \right) = \lim\limits_{n\to\infty} \frac{\partial h_n}{\partial x} = g$, contradicting $\|g - \frac{\partial f}{\partial x}\| > 0$. $\qquad\square$

**Proposition 1.** *Given any two functions $f : S \to \mathbb{R}$ and $g : S \to \mathbb{R}^d$ on $S \subseteq \mathbb{R}^d$ and a finite set $\Sigma \subset S$, there exists neural network $h$ with a ReLU (or a leaky ReLU) activation such that $\forall x \in \Sigma : f(x) = h(x)$ and $g(x) = \frac{\partial h}{\partial x}(x)$ (it has 0 training loss).*

*Proof.* We first prove the theorem in a special, 1-dimensional case (when $S$ is a subset of $\mathbb{R}$). Form now it will be assumed that $S$ is a subset of $\mathbb{R}$ and $\Sigma = \{\sigma_1 < \ldots < \sigma_n\}$ is a finite subset of $S$. Let $\varepsilon$ be smaller than $\frac{1}{5}\min(s_i - s_{i-1})$, $i = 2, \ldots, n$. We define a function $p_i$ as follows

$$p_i(x) = \begin{cases} \frac{f(\sigma_i) - g(\sigma_i)\varepsilon}{\varepsilon}(x - \sigma_i + 2\varepsilon) & \text{for} \quad x \in [\sigma_i - 2\varepsilon, \sigma_i - \varepsilon] \\ f(\sigma_i) + g(\sigma_i)(x - \sigma_i) & \text{for} \quad x \in [\sigma_i - \varepsilon, \sigma_i + \varepsilon] \\ -\frac{f(\sigma_i) + g(\sigma_i)\varepsilon}{\varepsilon}(x - \sigma_i - 2\varepsilon) & \text{for} \quad x \in [\sigma_i + \varepsilon, \sigma_i + 2\varepsilon] \\ 0 & \text{otherwise} \end{cases}.$$

Note that the functions $p_i$ have disjoint supports for $i \neq j$. We define $h(x) = \sum_{i=1}^n p_i(x)$. By construction, it has all the desired properties.

Now let us move to the general case, when $S$ is a subset of $\mathbb{R}^d$. We will denote by $\pi_k$ a projection of a $d$-dimensional point $\sigma$ onto the $k$-th coordinate. The obstacle to repeating the 1-dimensional proof in a straightforward matter (coordinate-by-coordinate) is that two or more of the points $\sigma_i$ can have one or more coordinates equal. We will use a linear change of coordinates to get past this technical obstacle. Let $A \in GL(d, \mathbb{R})$ be matrix such that there holds $\pi_k(A\sigma_i) \neq \pi_k(A\sigma_j)$ for any $i \neq j$ and any $K = 1, \ldots, d$. Such $A$ exists, as every condition $\pi_k(A\sigma_i) = \pi_k(A\sigma_j)$ defines a codimension-one submanifold in the space $GL(d, \mathbb{R})$, thus the complement of the union of all such submanifolds is a full dimension (and thus nonempty) subset of $GL(d, \mathbb{R})$. Using the one-dimensional construction we define functions $p^k(x)$, $k = 1, \ldots, d$, such that $p^k(\pi_k(A\sigma_i)) = \frac{1}{d}f(\sigma_i)$ and $(p^k)'(\pi_k(A\sigma_i)) = 0$. Similarly, we construct $q^k(x)$ in such manner $q^k(\pi_k(A\sigma_i)) = 0$ and $(q^k)'(\pi_k(A\sigma_i)) = A^{-1}g(\sigma_i)$. Note that those definitions a are valid because $\pi_k(A\sigma_i) \neq \pi_k(A\sigma_j)$ for $i \neq j$, so the right sides are well-defined unique numbers.

It remains to put all the elements together. This is done as follows. First we extend $p^k$, $q^k$ to the whole space $\mathbb{R}$ "trivially", i.e. for any $\mathbf{x} \in \mathbb{R}$, $\mathbf{x} = (x^1, \ldots, x^d)$ we define $P^k(\mathbf{x}) := p^k(x^k)$. Similarly, $Q_i^k(\mathbf{x}) := q_i^k(x^k)$. Finally, $h(\mathbf{x}) := \sum_{k=1}^d P^k(A\mathbf{x}) + \sum_{k=1}^d Q^k(A\mathbf{x})$. This function has the desired properties. Indeed for every $\sigma_i$ we have

$$h(\sigma_i) = \sum_{k=1}^d P^k(A\sigma_i) + \sum_{k=1}^d Q^k(A\sigma_i) = \sum_{k=1}^d p^k(\pi_k(A\sigma_i)) + \sum_{k=1}^d 0 = f(\sigma_i)$$

and

$$\frac{\partial h}{\partial x}(\sigma_i) = \sum_{k=1}^d (P^k)'(A\sigma_i) + \sum_{k=1}^d (Q^k)'(A\sigma_i) = \sum_{k=1}^d 0 + \sum_{k=1}^d \frac{\partial Q^k}{\partial x}(\pi_k(A\sigma_i)) =$$

$$A\sum_{k=1}^d (0, \ldots, \underbrace{(q^k)'(\pi_k(A\sigma_i))}_{k}, \ldots, 0)^T = A \cdot A^{-1}g(\sigma_i) = g(\sigma_i).$$

$$\square$$

This completes the proof.

**Proposition 3.** *There holds $K_{sob}(\mathcal{F}_{\mathrm{G}}) < K_{reg}(\mathcal{F}_{\mathrm{G}})$ and $K_{sob}(\mathcal{F}_{\mathrm{PL}}) < K_{reg}(\mathcal{F}_{\mathrm{PL}})$.*

*Proof.* Gaussian PDF functions form a 2-parameter family $\frac{1}{\sqrt{2\pi\sigma^2}}e^{-\frac{(x-\mu)^2}{2\sigma^2}}$. Therefore, determining $f$ in that family is equivalent to determining the values of $\mu$ and $\sigma^2$. Given $\alpha = \frac{1}{\sqrt{2\pi\sigma^2}}e^{-\frac{(x-\mu)^2}{2\sigma^2}}$, $\beta = -\frac{x-\mu}{\sigma^2\sqrt{2\pi\sigma^2}}e^{-\frac{(x-\mu)^2}{2\sigma^2}}$, we get $\frac{\beta}{\alpha} = -\frac{x-\mu}{\sigma^2}$ and $2\ln(\sqrt{2\pi}\alpha) = -\ln(\sigma^2) - \frac{(x-\mu)^2}{\sigma^2}$. Thus $2\ln(\sqrt{2\pi}\alpha) =$

$-\ln(\sigma^2) - \frac{\beta^2}{\alpha^2}\sigma^2$. The right hand side is a strictly decreasing function of $\sigma^2$. Substituting its unique solution to $\frac{\beta}{\alpha} = -\frac{x-\mu}{\sigma^2}$ we determine $\mu$. Thus $K_{sob}$ is equal to 1 for the family of Gaussian PDF functions.

On the other hand, there holds $K_{reg} > 2$ for the family of Gaussian PDF functions. For example, $N(2,1)$ and $N(2.847..., 1.641...)$ have the same values at $x = 0$ and $x = 3$ (existence of a "real" solution near this approximate solution is an immediate consequence of the Implicit Function Theorem). This ends the proof for the $\mathcal{F}_{\mathrm{G}}$ family

We will discuss the family $\mathcal{F}_{\mathrm{PL}}$ now. Every linear function is uniquely determined by its value at a single point and its derivative. Thus, for any function $f \in \mathcal{F}_{\mathrm{PL}}$, as the partition $D = D_1 \cup \ldots \cup D_n$ is fixed, it is sufficient to know the values and the values of the derivative of $f$ in $\sigma_1 \in D_n, \ldots, \sigma_1 \in D_n$ to determine it uniquely. On the other hand, we need at least $d + 1$ (recall that $d$ is the dimension of the domain of $f$) in each of the domains $D_i$ to determine $f$ uniquely, if we are allowed to look only at the values.

<div align="right">□</div>

**Proposition 4.** *If an approximator $SG(h, y|\theta)$ produces a valid gradient vector field of some scalar function $L$ then approximator's Jacobian matrix has to be symmetric.*

*Proof.* This comes directly from the fact that order of differentiation does not matter, so if we assume that there exists $L$ such that $SG(h, y|\theta) = \frac{\partial L}{\partial h}$ then

$$\forall_{ij} \quad \mathrm{Jac}(SG)_{ij} = \frac{\partial SG(h,y|\theta)_i}{\partial h_j} = \frac{\partial^2 L}{\partial h_i \partial h_j} = \frac{\partial^2 L)}{\partial h_j \partial h_i} = \frac{\partial SG(h,y|\theta)_j}{\partial h_i} = \mathrm{Jac}(SG)_{ji}$$

<div align="right">□</div>

## 2   Artificial Datasets

Figure 5: Ackley function (on the left) and its models using regular neural network training (left part of each plot) and Sobolev Training (right part). We also plot the vector field of the gradients of each predictor underneath the function plot.

Functions used (visualised at Figures 5-11):

- Ackley's

$$f(x,y) = -20\exp\left(-0.2\sqrt{0.5(x^2+y^2)}\right) - \exp\left(0.5(\cos(2\pi x) + \cos(2\pi y))\right) + e + 20,$$

for $x, y \in [-5, 5] \times [-5, 5]$

- Beale's

$$f(x,y) = (1.5 - x + xy)^2 + (2.25 - x + xy^2)^2 + (2.625 - x + xy^3)^2,$$

for $x, y \in [-4.5, 4.5] \times [-4.5, 4.5]$

- Booth

$$f(x,y) = (x + 2y - 7)^2 + (2x + y - 5)^2,$$

for $x, y \in [-10, 10] \times [-10, 10]$

<div align="center">13</div>

Figure 6: Beale function (on the left) and its models using regular neural network training (left part of each plot) and Sobolev Training (right part). We also plot the vector field of the gradients of each predictor underneath the function plot.

Figure 7: Booth function (on the left) and its models using regular neural network training (left part of each plot) and Sobolev Training (right part). We also plot the vector field of the gradients of each predictor underneath the function plot.

- Bukin

$$f(x, y) = 100\sqrt{|y = 0.01x^2|} + 0.01|x + 10|,$$

for $x, y \in [-15, -5] \times [-3, 3]$

- McCormick

$$f(x, y) = \sin(x + y) + (x - y)^2 - 1.5x + 2.5y + 1,$$

for $x, y \in [-1.5, 4] \times [-3, 4]$

- Rosenbrock

$$f(x, y) = 100(y - x^2)^2 + (x - 1)^2,$$

for $x, y \in [-2, 2] \times [-2, 2]$

- Styblinski-Tang

$$f(x, y) = 0.5(x^4 - 16x^2 + 5x + y^4 - 16y^2 + 5y),$$

for $x, y \in [-5, 5] \times [-5, 5]$

Networks were trained using the Adam optimiser with learning rate $3e - 5$. Training set has been sampled uniformly from the domain provided. Test set consists always of 10,000 points sampled uniformly from the same domain.

Figure 8: Bukin function (on the left) and its models using regular neural network training (left part of each plot) and Sobolev Training (right part). We also plot the vector field of the gradients of each predictor underneath the function plot.

Figure 9: McCormick function (on the left) and its models using regular neural network training (left part of each plot) and Sobolev Training (right part). We also plot the vector field of the gradients of each predictor underneath the function plot.

# 3 Policy Distillation

Agents policies are feed forward networks consisting of:

- 32 8x8 kernels with stride 4
- ReLU nonlinearity
- 64 4x4 kernels with stride 2
- ReLU nonlinearity
- 64 3x3 kernels with stride 1
- ReLU nonlinearity
- Linear layer with 512 units
- ReLU nonlinearity
- Linear layer with 3 (Pong), 4 (Breakout) or 6 outputs (Space Invaders)
- Softmax

They were trained with A3C [16] over 80e6 steps, using history of length 4, greyscaled input, and action repeat 4. Observations were scaled down to 84x84 pixels.

Figure 10: Rosenbrock function (on the left) and its models using regular neural network training (left part of each plot) and Sobolev Training (right part). We also plot the vector field of the gradients of each predictor underneath the function plot.

Figure 11: Styblinski-Tang function (on the left) and its models using regular neural network training (left part of each plot) and Sobolev Training (right part). We also plot the vector field of the gradients of each predictor underneath the function plot.

Data has been gathered by running trained policy to gather 100K frames (thus for 400K actual steps). Split into train and test sets has been done time-wise, ensuring that test frames come from different episodes than the training ones.

Distillation network consists of:

- 16 8x8 kernels with stride 4
- ReLU nonlinearity
- 32 4x4 kernels with stride 2
- ReLU nonlinearity
- Linear layer with 256 units
- ReLU nonlinearity
- Linear layer with 3 (Pong), 4 (Breakout) or 6 outputs (Space Invaders)
- Softmax

and was trained using Adam optimiser with learning rate fitted independently per game and per approach between $1e - 3$ and $1e - 5$. Batch size is 200 frames, randomly selected from the training set.

# 4 Synthetic Gradients

All models were trained using multi-GPU optimisation, with Sync main network updates and Hogwild SG module updates.

## 4.1 Meaning of Sobolev losses for synthetic gradients

In the setting considered, the true label $y$ is used only as a conditioning, however one could also provide supervision for $\partial m(h, y|\theta)/\partial y$. So what is the actual effect this Sobolev losses have on SG estimator? For $L$ being log loss, it is easy to show, that they are additional penalties on matching $\log p(h, y)$ to $\log p_h$, namely:

$$\|\partial m(h, y|\theta)/\partial y - \partial L(h, y)/\partial y\|^2 = \|\log p(h|\theta) - \log p_h\|^2$$

$$\|m(h, y|\theta) - L(h, y)\|^2 = (\log p(h|\theta)_{\hat{y}} - \log p_{h\hat{y}})^2,$$

where $\hat{y}$ is the index of "1" in the one-hot encoded label vector $y$. Consequently loss supervision makes sure that the internal prediction $\log p(h|\theta)$ for the true label $\hat{y}$ is close to the current prediction of the whole model $\log p_h$. On the other hand matching partial derivatives wrt. to label makes sure that predictions for all the classes are close to each other. Finally if we use both – we get a weighted sum, where penalty for deviating from the prediction on the true label is more expensive, than on all remaining ones[6].

## 4.2 Cifar10

All Cifar10 experiments use a deep convolutional network of following structure:

- 64 3x3 kernels with stride 1
- BatchNorm and ReLU nonlinearity
- 64 3x3 kernels with stride 1
- BatchNorm and ReLU nonlinearity
- 128 3x3 kernels with stride 2
- BatchNorm and ReLU nonlinearity
- 128 3x3 kernels with stride 1
- BatchNorm and ReLU nonlinearity
- 128 3x3 kernels with stride 1
- BatchNorm and ReLU nonlinearity
- 256 3x3 kernels with stride 2
- BatchNorm and ReLU nonlinearity
- 256 3x3 kernels with stride 1
- BatchNorm and ReLU nonlinearity
- 256 3x3 kernels with stride 1
- BatchNorm and ReLU nonlinearity
- 512 3x3 kernels with stride 2
- BatchNorm and ReLU nonlinearity
- 512 3x3 kernels with stride 1
- BatchNorm and ReLU nonlinearity
- 512 3x3 kernels with stride 1
- BatchNorm and ReLU nonlinearity
- Linear layer with 10 outputs
- Softmax

with L2 regularisation of $1e-4$. The network is trained in an asynchronous manner, using 10 GPUs in parallel. Each worker uses batch size of 32. The main optimiser is Stochastic Gradient Descent with momentm of 0.9. The learning rate is initialised to 0.1 and then dropped by an order of magniture after 40K, 60K and finally after 80K updates.

Each of the three SG modules is a convolutional network consisting of:

- 128 3x3 kernels with stride 1
- ReLU nonlinearity
- Linear layer with 10 outputs
- Softmax

It is trained using the Adam optimiser with learning rate $1e-4$, no learning rate schedule is applied. Updates of the synthetic gradient module are performed in a Hogwild manner. Losses used for both loss prediction and gradient estimation are L1.

For direct SG model we used architecture described in the original paper – 3 resolution preserving layers of 128 kernels of 3x3 convolutions with ReLU activations in between. The only difference is that we use L1 penalty instead of L2 as empirically we found it working better for the tasks considered.

## 4.3   Imagenet

All ImageNet experiments use ResNet50 network with L2 regularisation of $1e-4$. The network is trained in an asynchronous manner, using 34 GPUs in parallel. Each worker uses batch size of 32. The main optimiser is Stochastic Gradient Descent with momentum of 0.9. The learning rate is initialised to 0.1 and then dropped by an order of magnitude after 100K, 150K and finally after 175K updates.

The SG module is a convolutional network, attached after second ResNet block, consisting of:

- 64 3x3 kernels with stride 1
- ReLU nonlinearity
- 64 3x3 kernels with stride 2
- ReLU nonlinearity
- Global averaging
- 1000 1x1 kernels
- Softmax

It is trained using the Adam optimiser with learning rate $1e-4$, no learning rate schedule is applied. Updates of the synthetic gradient module are performed in a Hogwild manner. Sobolev losses are set to L1.

Regular data augmentation has been applied during training, taken from the original Inception V1 paper.

## 5   Gradient-based attention transfer

Zagoruyko et al. [31] recently proposed a following cost for transfering attention model $f$ to model $g$ parametrised with $\theta$, under the cost $L$:

$$L_{\text{transfer}}(\theta) = L(g(x|\theta)) + \alpha \|\partial L(g(x|\theta))/\partial x - \partial L(f(x))/\partial x\|_2 \tag{3}$$

where the first term simply is the original minimisation problem, and the other measures loss sensitivity of the target ($f$) and tries to match the corresponding quantity in the model $g$. This can be seen as a Sobolev training under four additional assumptions:

1. ones does not model $f$, but rather $L(f(x))$ (similarly to our Synthetic Gradient model – one constructs loss predictor),
2. $L(f(x)) = 0$ (target model is perfect),
3. loss being estimated is non-negative ($L(\cdot) \geq 0$)
4. loss used to measure difference in predictor values (loss estimates) is $L_1$.

If we combine these four assumptions we get

$$L_{\text{sobolev}}(\theta) = \|L(g(x|\theta)) - L(f(x))\|_1 + \alpha\|\partial L(g(x|\theta))/\partial x - \partial L(f(x))/\partial x\|_2$$
$$= \|L(g(x|\theta))\|_1 + \alpha\|\partial L(g(x|\theta))/\partial x - \partial L(f(x))/\partial x\|_2$$
$$= L(g(x|\theta)) + \alpha\|\partial L(g(x|\theta))/\partial x - \partial L(f(x))/\partial x\|_2.$$

Note, however than in general these approaches are not the same, but rather share the idea of matching gradients of a predictor and a target in order to build a better model.

In other words, Sobolev training exploits derivatives to find a closer fit to the target function, while the transfer loss proposed adds a sensitivity-matching term to the original minimisation problem instead. Following observation make this distinction more formal.

**Remark 2.** *Lets assume that a target function $L \circ f$ belongs to hypotheses space $\mathcal{H}$, meaning that there exists $\theta_f$ such that $L(g(\cdot|\theta_f)) = L(f(\cdot))$. Then $\theta_f$ is a minimiser of Sobolev loss, but does not have to be a minimiser of transfer loss defined in Eq. (3).*

*Proof.* By the definition of Sobolev loss it is non-negative, thus it suffices to show that $L_{\text{sobolev}}(\theta_f) = 0$, but

$$L_{\text{sobolev}}(\theta_f) = \|L(g(x|\theta_f)) - L(f(x))\| + \alpha\|\partial L(g(x|\theta_f))/\partial x - \partial L(f(x))/\partial x\|$$
$$= \|L(f(x)) - L(f(x))\| + \alpha\|\partial L(f(x))/\partial x - \partial L(f(x))/\partial x\| = 0.$$

By the same argument we get for the transfer loss

$$L_{\text{transfer}}(\theta_f) = L(g(x|\theta_f)) + \alpha\|\partial L(g(x|\theta_f))/\partial x - \partial L(f(x))/\partial x\|$$
$$= L(g(x|\theta_f)) + \alpha\|\partial L(f(x))/\partial x - \partial L(f(x))/\partial x\| = L(g(x|\theta_f)).$$

Consequently, if there exists another $\theta_h$ such that $L(g(x|\theta_h)) < L(g(x|\theta_f)) - \alpha\|\partial L(g(x|\theta_h))/\partial x - \partial L(f(x))/\partial x\|$, then $\theta_f$ is not a minimiser of the loss considered.

To show that this final constraint does not lead to an empty set, lets consider a class of constant functions $g(x|\theta) = \theta$, and $L(p) = \|p\|^2$. Lets fix some $\theta_f > 0$ that identifies $f$, and we get:

$$L_{\text{transfer}}(\theta_f) = L(g(x|\theta_f)) = \theta_f^2 > 0$$

and at the same time for any $|\theta_h| < \theta_f$ (i.e. $\theta_h = \theta_f/2$) we have:

$$L_{\text{transfer}}(\theta_h) = L(g(x|\theta_h)) + \alpha\|\partial L(g(x|\theta_h))/\partial x - \partial L(g(x|\theta_f))/\partial x\|$$
$$= \theta_h^2 + \alpha(0 - 0) = \theta_h^2 < \theta_f^2 = L_{\text{transfer}}(\theta_f).$$

$\square$

## Footnotes

[6]Adding $\partial L/\partial y$ supervision on toy MNIST experiments increased convergence speed and stability, however due to TensorFlow currently not supporting differentiating cross entropy wrt. to labels, it was omitted in our large-scale experiments.