[Reviews · NeurIPS 2017]

Reviewer 1



I have enjoyed reading this paper. The idea is clear and very intuitive. Figure 1 is very well conceived. Now, I must say that, throughout my reading, I was looking for the authors to "confess" that the first-order derivatives might be entirely useless in certain situations. But in a situation where both f(x) and f'(x) are given, taken as ground truth, then it is certainly a wise idea to use the information about f'(x) in order to train a new network that approximates f(x) to solve a task. However the problem is that, when a certain model M1 is trained to minimize a loss on a training set, then there is nothing that guarantees the sanity/usefulness of the gradients of M1. The derivatives depend a lot on the activation functions being used, for example. Then I don't see at all why there would be any general principle that would encourage us to imitate those gradients in order to learn another model M2 based on M1. It is very possible that this doesn't happen in practice, on common datasets and reasonable models, and that's interesting. I think that it's partly what the paper studies : does this thing work in practice ? Okay, great ! Worth discussing in a paper with good experiments. Accept ! In the past, I have had a conversation with one of the great ML authorities in which he/she casually claimed that when two functions (f1, f2) are close, then their derivatives are also close. This doesn't sound unreasonable as a mental shortcut, but it's ABSOLUTELY INCORRECT in the mathematical sense. Yet people use that shortcut. The authors refer to this indirectly, but I would really like to see the authors be more explicit about that fact, and I think it would make the paper slightly better. Just to be sure that nobody makes that mental shortcut when they read this stimulating paper. I had a bit more difficulty understanding the subtleties of the Synthetic Gradient portion of the paper. I trust that it's sound. Also, in Figure 4 I'm looking at certain plots and it seems like "regular distillation" very often gets worse with more iterations. (The x-axis are iterations, right?) Sobolev training looks better, but it's hard to say if it's just because the hyperparameters were not selected correctly. They kinda make regular distillation look bad. Moreover, the y-axis really plays with the scales and makes the differences look way bigger than they are. It's hard to know how meaningful those differences are. To sum it up, I'm glad that I got to read this paper. I would like to see better, and more meaningful experiments, in a later paper, but I'm happy about what I see now.

Reviewer 2



This paper proposes a new kind of objective function to train neural networks when the gradient of f(x) w.r.t. x is known, called Sobolev training. In addition to providing some theory on sample complexity, the paper provides meaningful empirical evidence of their method. Sobolev Training consists in training some model m(x;theta) to not only match f(x) but also the first Kth order derivatives of the output w.r.t to the input, i.e. train all grad^k_x m(x) to match grad^k_x f(x). This paper is very well structured: it proposes an idea, motivates some nice theoretical properties behind it, then demonstrates its utility in some machine learning scenarios. It's clearly written and everything seems to make sense. One downside of this paper is that it doesn't spend too much time motivating its theoretical approach. The polynomial approximation example is a good way to motivate this in 2D, but does this intuition hold when dealing with neural networks? Evidently something good is going on empirically, but looking at figure 4, I'm wondering if "sample efficiency" is the right way to look at this (unless zooming in early in these curves would reveal something else).